# Sensitivity of Eurasian Rear-Edge Birch Populations to Regional Climate and Local Hydrological Conditions

Ester González de Andrés [1], Michele Colangelo [1,2], Reyes Luelmo-Lautenschlaeger [3], José Antonio López-Sáez [4] and Jesús Julio Camarero [1,*]

[1] Instituto Pirenaico de Ecología (CSIC), Avda. Montañana 1005, 50009 Zaragoza, Spain; ester.gonzalez@ipe.csic.es (E.G.d.A.)
[2] Scuola di Scienze Agrarie, Forestali, Alimentari, e Ambientali, Università della Basilicata, 85100 Potenza, Italy
[3] Institut des Sciences de l'Évolution de Montpellier, CNRS, Université de Montpellier, 34394 Montpellier, France
[4] Research Group Environmental Archaeology, Instituto de Historia (CSIC), 28037 Madrid, Spain
[*] Correspondence: jjcamarero@ipe.csic.es; Tel.: +34-976-369-393

**Abstract:** South rear-edge populations of widely distributed temperate and boreal tree species such as birches (*Betula pubescens* and *Betula pendula*) are considered particularly vulnerable to climate warming, and at the same time, they constitute genetic reservoirs of drought-adapted ecotypes. Here, we compared radial growth patterns and responses to climate, river, or reservoir flows and a drought index of rear-edge (southernmost) populations (Toledo Mountains, central-southern Spain) with populations located in northern Spain of *B. pubescens* and *B. pendula*. Then, we performed a comparative analysis across Europe of *B. pendula* populations. The main climatic constraint of birch growth was a high summer water deficit, although the effect of local hydrological conditions was particularly important in rear-edge populations. We found declining growth trends in rear-edge stands dating from the early 21st century, related to decreasing water availability and increasing aridity. Our results also suggested distinct growth patterns and climate-growth associations of *B. pendula* across Europe that show how populations further south and in warmer locations were more sensitive to drought stress. Drought-induced growth decline can be exacerbated by local human land uses, leading to reduced river inflow, thus endangering birch populations at their southern distribution limit. Protection of threatened rear-edge birch populations requires adequate management of local water resources.

**Keywords:** *Betula pendula*; *Betula pubescens*; drought; growth decline; land use; peatland; Toledo Mountains

## 1. Introduction

Birch species (*Betula* L.) are deciduous, shade-intolerant shrubs and trees native to the northern hemisphere and widely distributed across boreal and temperate forests of Eurasia and North America [1]. Although birches have a wide ecological range, they are usually found in cool-wet sites with abundant soil water availability, such as mires, riverbanks, or in alpine or arctic treelines [2]. Birches play an important role as major pioneer species due to their efficient colonization of disturbed sites, increasing soil fertility and facilitating the establishment of other species [3,4], and due to their high growth rates at a young age [5,6]. So, birch species are valuable in natural or anthropogenic regeneration and contribute to the stability of forest ecosystems [7]. Birches are also important for maintaining biodiversity by supporting a high number of invertebrates [8], birds [9], and lichen species [10]. Moreover, they yield high-quality timber that is highly suitable for plywood production [3]. Therefore, birches have an important ecological and economic value.

However, ongoing climate change is leading to warming temperatures and increasing the frequency of extreme climate events, such as prolonged dry spells and heatwaves [11]. This has amplified drought stress and related forest disturbances during

the last decades [12]. As a consequence, forest dieback and growth decline events have been reported in multiple biomes [13,14]. These drought-induced dieback and mortality episodes have also been reported in birch forests of southeastern Siberia [15] and in the Himalayas [16]. Those studies emphasized that regional climate conditions caused the dieback, but local topography also amplified the negative impacts of soil moisture deficit [15,16]. However, we still lack adequate assessments of local vs. regional impacts on birch growth and decline.

In Europe, two main birch tree species are generally recognized: downy birch (*Betula pubescens* Ehrh.) and silver birch (*Betula pendula* Roth.), which have some of the largest geographical distribution areas in the continent [1,17]. However, in many parts of Europe, they are sympatric and can naturally hybridize, generating plants with intermediate morphological traits [18]. Both species are sensitive to water shortage in summer and snow regime in winter [19–22], as well as vulnerable to low temperatures in spring [23]. Nonetheless, there are some differences in their ecological requirements as *B. pubescens* is more suitable for cooler climates and wetter soils, whereas *B. pendula* has been shown to be more tolerant to drought and warmer temperatures [24,25]. In addition, nonlinear responses to environmental and climatic gradients have been described in birches [21,26,27].

Birch populations in southern Spain represent the southwestern distribution limit (rear edge) of both *B. pubescens* and *B. pendula* in Eurasia (Figure 1) [17]. These rear-edge populations are highly fragmented and show a strong genetic differentiation [28]. They mainly grow in particularly humid locations (e.g., near rivers, streams, or mires) within Mediterranean environments subjected to pronounced summer drought [29]. Palaeoecological studies show that birch populations in central-southern Spain (Toledo Mountains) survived past dry periods but showed a clear demise due to increasing livestock pressure [30,31]. According to these studies, ungulate densities and herbivory pressure reached unprecedented values during the last decades. The resulting heavy ungulate browsing that menaces birch recruitment, along with wildfire occurrence and habitat fragmentation, have been identified as major threats to those Spanish rear-edge birch populations [29,31].

Long-term isolation of rear-edge populations has often resulted in local adaptation, especially to increase the species' survival under increasingly dry and warm conditions but reducing plasticity, thus limiting their adaptability to rapid environmental changes [32–34]. Rear-edge populations have also been shown to be particularly vulnerable to climate warming despite their genetic adaptations [35,36]. Indeed, pioneer birch species such as *B. pendula* are forecasted to be among the most endangered by climate warming during the 21st century in some regions of southern Europe, where they may become locally extinct as their range shifts northwards or upwards according to species distribution models [37]. Therefore, analyses of the performance of rear-edge birch stands' growth responses to increasing aridification and alteration of local hydrological conditions are needed to better understand their vulnerability compared to more central populations.

Here, we studied radial growth patterns of rear-edge *B. pubescens* and *B. pendula* populations located in central-southern Spain, and we analyzed the differences with more central populations of the species at different spatial scales. For this purpose, we used a dendrochronological, retrospective approach based on tree-ring width data, which is an efficient proxy used for the long-term assessment of the responses of forests to environmental changes [38,39]. Our objectives were (i) to compare radial growth trends and growth responses to regional climate and local hydrological conditions between *B. pubescens* and *B. pendula* populations in central-southern Spain (Toledo Mountains) and northern Spain (Iberian System) and (ii) to address growth variability and climate-growth relationships of *B. pendula* across Europe. We expect that the radial growth of rear-edge populations would be more limited by low soil moisture than northern populations of Spain and Europe and that local changes in water supply determined by site-specific hydrology would modulate or even surpass regional climate effects on the tree growth of rear-edge populations.

## 2. Materials and Methods

### 2.1. Study Sites and Species

Firstly, we compared Spanish birch populations located in the Toledo Mountains and the Iberian System, where rear-edge and more central populations, respectively, of both species are found (Figure 1). The Toledo Mountains ("Montes de Toledo") study area is located in central-southern Spain, where several rear-edge birch populations are located [29]. There, we sampled two birch populations at the southern distribution limit of the species inhabiting sites with contrasting ecological conditions (Figure 1, Table 1). La Ventilla site is located within the "Cabañeros" National Park, and the downy birch (*Betula pubescens* Ehrh. subsp. *celtiberica* (Rothm. and Vasc.) Rivas Mart.) trees grow over waterlogged soils in a mire (Figure A1a), close to the Bullaque River. This is the only population of this species in the study area, which, according to palaeoecological data, was more widely distributed in the past [29,40]. The Riofrío site is found along the banks of a stream (Figure A1b), which is a tributary of the Bullaque River. Here, riparian forests are formed along a ca. 2 km long stream, and silver birch (*B. pendula* Roth. subsp. *fontqueri* (Roth.) G. Moreno and Peinado) is restricted to the riverbank. Understory vegetation of both stands included *Pteridium aquilinum*, *Ruscus aculeatus*, *Erica arborea*, *Cistus* spp., *Pistacia lentiscus*, and *Pistacia terebinthus*, among others. The study area in the Iberian System is found in northern Spain (Razoncillo Valley), where populations are found within or near a continuous species distribution (Figure 1) [17]. There, we sampled one *B. pubescens* population located in a rocky area. Among the accompanying species, we found *Fagus sylvatica* and *Salix* spp., and the understory was dominated by *Vaccinium myrtillus*, *Calluna vulgaris*, and *Erica* spp. Finally, information on a *B. pendula* population in the Iberian System (ES01 site) was obtained from the GenTree Database [41], which was located very close to the previous one (Table 1; Figure 1). Both populations are located close to a tributary stream of the Revinuesa River. It should be noted that the ES01 chronology finishes in 2015, although for the sake of simplicity, the period of analysis of the Spanish birch populations will be referred to as 1980–2021.

**Table 1.** Characteristics (mean ± standard error) of sampled sites and birch trees.

| | *Betula pubescens* | | *Betula pendula* | |
| --- | --- | --- | --- | --- |
| **Site ID** | **La Ventilla** | **Razoncillo** | **Riofrío** | **ES01** |
| Latitude (N) | 39°20′25″ | 41°59′12″ | 39°05′00″ | 41°58′08″ |
| Longitude (W) | 4°16′40″ | 2°38′38″ | 4°30′10″ | 2°37′19″ |
| Elevation (m a.s.l.) | 638 | 1555 | 648 | 1290 |
| DBH (cm) | 49.9 ± 3.1 | 35.0 ± 1.3 | 40.8 ± 1.6 | 29.8 ± 1.8 |
| Tree age (years) | 38 ± 2 | 67 ± 4 | 46 ± 2 | 40 ± 2 |
| No. sampled trees (No. radii) | 21 (44) | 12 (24) | 23 (46) | 25 (50) |
| TRW (mm) [1] | 5.01 ± 0.27 | 1.43 ± 0.11 | 3.10 ± 0.22 | 3.46 ± 0.28 |
| AC [1] | 0.71 ± 0.03 | 0.62 ± 0.04 | 0.73 ± 0.02 | 0.52 ± 0.06 |
| *Rbar* [1] | 0.278 | 0.360 | 0.296 | 0.239 |
| MS [1] | 0.297 | 0.369 | 0.288 | 0.318 |
| EPS [1] | 0.864 | 0.871 | 0.906 | 0.863 |

Variables' abbreviations: diameter at breast height (DBH), tree-ring width (TRW), first-order autocorrelation (AC), mean inter-series correlation (*Rbar*), mean sensitivity (MS), Expressed Population Signal (EPS). [1] Calculated for the period 1980–2021 (with EPS > 0.85) on raw (TRW, AC) or standardized (*Rbar*, MS) ring-width values.

Secondly, we compiled tree-ring width data of *B. pendula* across Europe, merging our own field data and the GenTree Database [41]. The authors of this dataset provided individual tree-ring width data, along with tree- and site-level characteristics. In total, growth variability and climate responses of 23 *B. pendula* populations were compared across a wide environmental gradient throughout Europe (Figure 1). Information on the site and tree characteristics can be found in Table A1, while methodological issues are specified in Martínez-Sancho et al. [41].

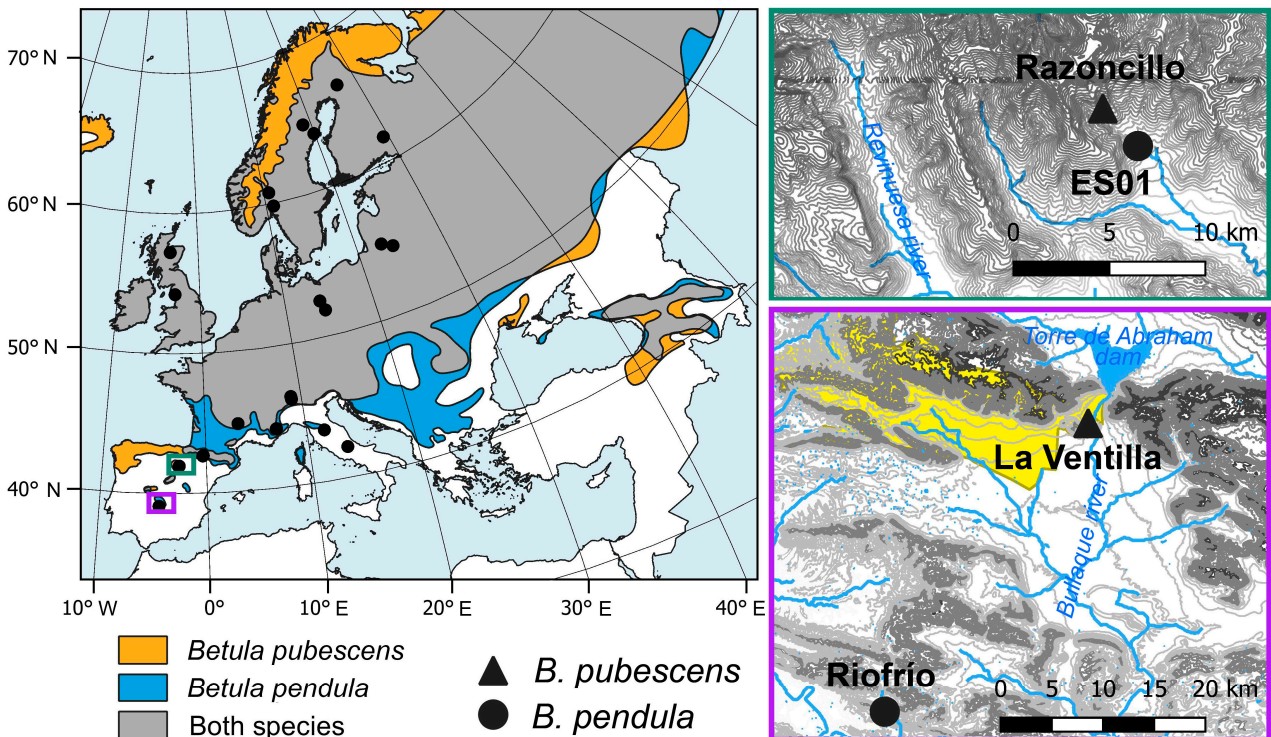

**Figure 1.** Distribution range of *Betula pubescens* (orange area), *B. pendula* (blue area), and the area where the distribution ranges of both species overlap (grey area) according to de Rigo et al. [42] (**left**). Location of Spanish study sites corresponding to the Iberian System (**top right**) and the Toledo Mountains (**bottom right**). The yellowish area indicates the limits of the "Cabañeros" National Park. Symbols indicate the location of the study sites of *B. pubescens* (triangles) and *B. pendula* (circles).

### 2.2. Climate and River Flow Data

Long-term and homogeneous records of climatic data from each study site were retrieved from the 0.1°-gridded E-OBS v 27.0e dataset [43]. To characterize drought severity, we calculated the Standardized Precipitation-Evapotranspiration Index (SPEI) for the period 1980–2021. The SPEI is a standardized multi-scalar drought index based on the accumulated water deficit, in which negative values indicate a negative cumulative water balance, i.e., the difference between precipitation and potential evapotranspiration (P-PET) [44]. SPEI values from January to December at different time scales were calculated using the package SPEI [45] in R software [46].

On the one hand, study sites in Spain showed contrasting climatic regimes (Figure A1c). The climate in the Toledo Mountains is Mediterranean with a marked period of summer drought and spring and autumn rainfall, and in the Iberian System is continental Mediterranean with cold winters. The mean annual temperature during the 1980–2021 period was higher in the Toledo Mountains, with slight differences between La Ventilla (15.0 °C) and Riofrío (15.9 °C), than in the study sites of the Iberian System (9.2 °C). The coldest month was January, and the warmest month was July (Figure A1c). The mean annual precipitation was 410 mm in La Ventilla, 395 mm in Riofrío, and 600 mm in the Iberian System study sites. The driest months were July-August, whereas October-November and April-May were the wettest months in southern and northern sites, respectively (Figure A1c). Summer water balance is decreasing at Toledo Mountains (Figure A2).

On the other hand, the mean annual temperature among the different populations of *B. pendula* analyzed in the European assessment ranges from 0.8 °C in northern Finland (FI19) to 15.9 °C in central-southern Spain (ES03, Riofrío). Total annual precipitation had minimum values at Riofrío of 415 mm and maximum values at Switzerland study sites of almost 1500 mm. Concurrently, the annual water balance was highest in the southern

United Kingdom (GB13) and Switzerland and lowest at the rear-edge population of Riofrío (Table A1).

To characterize local hydrological conditions, we retrieved a series of monthly flow data from water streams near the Spanish study sites from the Spanish National Flow and Discharge Database ("Centro de Estudios y Experimentación de Obras Públicas", data available at the webpage https://ceh.cedex.es (accessed on 30 March 2023)). For the Toledo Mountains sites, we obtained data from the Bullaque River (Luciana, 38°35′28″ N, 4°10′16″ W, 536 m a.s.l.) covering the period 1965–2019. For the Iberian System sites, we obtained data from the Revinuesa River (Vinuesa, 41°54′47″ N, 2°45′23″ W, 1090 m a.s.l.) for the period 1962–2019. Additionally, we obtained reservoir data, outflows, and inflows from the "Torre de Abraham" reservoir (39°22′28″ N, 4°14′44″ W, 700 m a.s.l.), close to the Toledo Mountains sites, for the period 1978–2019. According to these data, the Bullaque River flow and "Torre de Abraham" reservoir inflows peak in January and February and reach minimum values from July to October. The pattern of the Revinuesa River flow was similar, although maximum values were delayed (March–April).

In La Ventilla, the mires are progressively drying due to the water extraction from the phreatic zone for agriculture in nearby intensively irrigated crop fields. Soil water extraction intensified in the 2010s and has led to a progressive decline in the birch population reducing growth and increasing mortality rates [47].

### 2.3. Field Sampling and Dendrochronological Methods

In total, 81 birch trees (33 *B. pubescens* and 48 *B. pendula*) were sampled at the Spanish sites. Two cores were extracted at 1.3 m height from each tree, perpendicular to the maximum slope, using 5 mm Pressler increment borers (Haglöf, Sweden) for dendrochronological analysis. Often birch trees were multi-stemmed so that the trunk with the largest DBH and most vertical was selected for core extraction. Cores were air-dried, glued onto wooden mounts, and sanded until tree rings were clearly visible [38]. All samples were visually cross-dated, and tree-ring width (TRW) was measured with a 0.001 mm resolution using scanned images (resolution 2400 dpi) and the CooRecorder and CDendro software [48]. The quality of cross-dating was checked using the COFECHA software, which calculates moving correlations between individual series of ring-width values and the mean sites chronologies [49].

The individual tree TRW series were transformed into the basal area increment (BAI) series because it is a two-dimensional measure of stem increment in an area that is known to better reflect the growth of the whole tree than the one-dimensional ring width [50]. BAI series were calculated using the following equation and assuming concentric rings:

$$\text{BAI} = \pi \, (\text{R}^2_{\,t} - \text{R}^2_{\,t-1}), \tag{1}$$

where $\text{R}^2_{\,t}$ and $\text{R}^2_{\,t-1}$ are the radii corresponding to the current (t) and prior (t − 1) years, respectively.

All individual TRW chronologies were detrended using a spline of two- third of the growth series length and a 0.5 response cut-off. A bi-weight robust mean was used to calculate mean site standardized chronologies of ring-width indices (RWI). We also calculated several dendrochronological statistics to characterize tree-ring width data [38]: the mean and first-order autocorrelation (AC) of TRW data, the mean inter-series correlation (*Rbar*) of individual series of ring-width indices, the mean sensitivity (MS) of standardized ring-width indices (a measure of relative change in width between consecutive rings), and the Expressed Population Signal (EPS) of ring-width indices which measures the coherence and replication of site chronologies.

### 2.4. Statistical Analyses

We used the Kruskal–Wallis test to evaluate differences in DBH, tree age, and mean TRW between Spanish sites within each species. Breakpoint analysis was applied to detect structural changes in the BAI series corresponding to the onset of growth decline.

Then, trends in radial growth before and after the breakpoint dates were assessed using the Kendall $\tau$ statistic. Climate–growth relationships were assessed by calculating bootstrapped correlations between site-standardized chronologies (RWI) and monthly mean temperature and precipitation sums for the 1980–2021 period. The window of analysis spanned from previous September to September of the year of tree-ring formation. We also calculated correlations between RWIs and SPEI, which were calculated at 1-, 3-, 6-, 9-, and 12-month time scales, to quantify the relationship between radial growth variability and water deficit. To detect potential instabilities between growth indices and selected climatic data, we computed moving bootstrapped correlations considering moving 20-year long intervals lagged by one year. Pearson correlations were calculated to evaluate the statistical relationship between site RWI series and river flow and reservoir data. The aggregate effect of regional climate and local hydrological conditions on radial growth was addressed by means of linear mixed-effects models (LMM; [51]). One model was adjusted for each site and species, including tree-level RWI as response variables and SPEI, temperature, precipitation, and river flow (or reservoir inflow) series as fixed effects. Selected variables were those showing significant correlations in the previous analysis. The presence of multicollinearity among predictors was tested using variance inflation factors (VIF), discarding variables with VIF > 3.

We used generalized additive mixed models (GAMM; [52]) to describe growth responses to climate of *B. pendula* populations across Europe. Predictors were included as smooth functions of temperature, precipitation, and drought index (SPEI) using thin-plate regression splines with a maximum of four degrees of freedom to allow for nonlinear responses [53]. We also included site as a random effect and a first-order autocorrelation structure. The selection of the climatic variables was based on minimizing the Akaike Information Criterion (AIC; [54]). We also calculated the marginal ($R^2$m) and conditional ($R^2$c) $R^2$ values, which account for the effects of fixed and fixed plus random factors, respectively.

To evaluate the relationships among European *B. pendula* RWI chronologies and compare them with the rear-edge population in central-southern Spain, we calculated a Principal Component Analysis (PCA) on the covariance matrix considering the common and best-replicated period 1985–2015. We kept the first (PC1) and second (PC2) principal components because they individually accounted for at least 15% of the variance. We also performed a PCA on the covariance matrix of site RWI–climate correlations to assess relationships in the growth response to temperature and precipitation among different European populations. Finally, we calculated correlations of PCA scores and loadings with geographical and climatic variables to decipher continental-scale patterns of *B. pendula* growth and climate sensitivity.

All statistical analyses were performed within the R software [46]. Processing of radial growth series and dendrochronological statistics were calculated using the R package *dplR* [55]. Trends in temporal series and breakpoints analysis were assessed with the R packages *Kendall* [56] and *strucchange* [57], respectively. Relationships between radial growth series and climate were assessed using the R package *treeclim* [58]. We used *lme4* and *lmerTest* packages to fit LMMs [59,60]. The GAMMs were fitted using the R package *mgvc* [61].

## 3. Results

### 3.1. Growth Patterns of Birch Populations in Southern and Northern Spain

The mean DBH of *B. pubescens* trees did not differ between Spanish study sites ($\chi^2 = 16.757$, $p = 0.159$), but *B. pendula* trees showed higher DBH in the southern population than in the northern population ($\chi^2 = 18.519$, $p < 0.001$) (Table 1). Tree age was significantly higher at the Razoncillo site compared to La Ventilla ($\chi^2 = 22.129$, $p = 0.036$), but no differences were found between *B. pendula* populations ($\chi^2 = 3.551$, $p = 0.159$). The oldest sampled individuals were around 90 years old at the northern stand of *B. pubescens*. Mean growth rates (TRW) were higher in La Ventilla than in Razoncillo ($\chi^2 = 23.541$, $p = 0.023$) (Figure 2), and similar findings were found between Riofrío and ES01 ($\chi^2 = 0.146$, $p = 0.703$). We found breakpoints in BAI series corresponding to 2007 (both *B. pubescens*

stands), 2001 (Riofrío), and 1984 (ES01). Before each breakpoint date, birch trees of every site showed a positive growth trend (La Ventilla: $\tau = 0.762$, $p < 0.001$; Razoncillo: $\tau = 0.216$, $p = 0.033$; Riofrío: $\tau = 0.267$, $p = 0.016$; ES01: $\tau = 0.446$, $p = 0.004$). Afterward, growth trends became negative in the southern populations with a stepper slope in La Ventilla ($\tau = -0.513$, $p = 0.017$) than in Riofrío ($\tau = -0.368$, $p = 0.025$), whereas non-significant and positive trends were found at the northern populations (Razoncillo: $\tau = -0.259$, $p = 0.199$; ES01: $\tau = 0.376$, $p = 0.003$) (Figure 2).

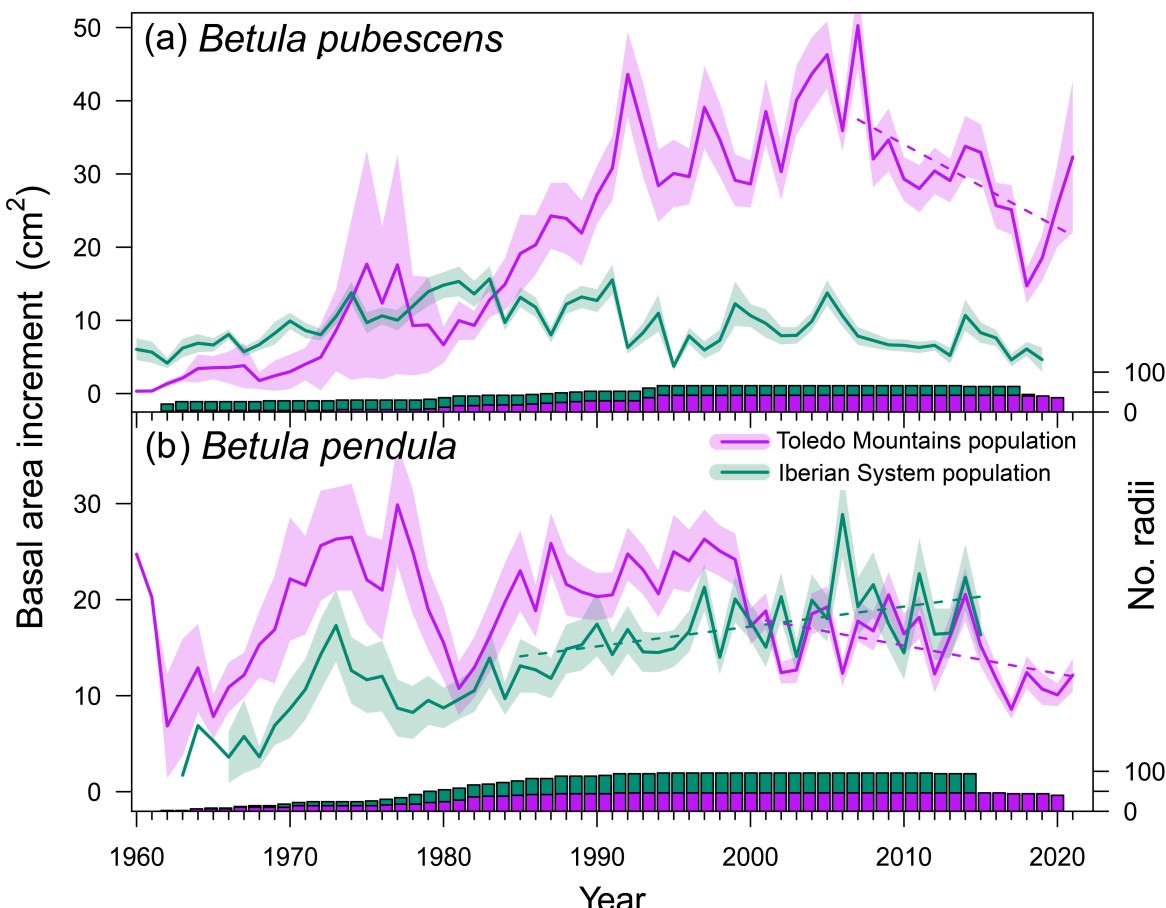

**Figure 2.** Variation of basal area increment (BAI) in the two study species, *Betula pubescens* (**a**) and *B. pendula* (**b**). Solid lines represent the means and shaded areas around them, the standard error of the mean. Dash lines show significant ($p < 0.05$) trends in mean BAI before and after detected breakpoint dates. The bars show sample depth (number of measured radii) with the same colors as those used for symbols.

### 3.2. Effect of Climate and Local Hydrlogy on the Growth of Spanish Birch Stands

Mean annual temperature showed positive trends in both Spanish study sites during the period 1980–2021, while no significant trends were found for mean annual precipitation. Summer water balance significantly decreased at Toledo Mountains, but the same pattern was not detected in the Iberian System (Figure A2). These results are in line with the negative trend of the annual flow of the Bullaque River found for the period 1980–2019 ($\tau = -0.275$, $p = 0.011$), which is not found in Revinuesa River ($\tau = -0.149$, $p = 0.114$).

Bootstrapped correlations showed that associations between site-level RWI chronologies and temperature were more common in northern populations of both birch species, whereas significant relationships with precipitation were more frequent in southern rearedge birch populations (Figure 3). In the Iberian System, birch trees positively responded to April, May (*B. pendula*), and June (*B. pubescens*) mean temperatures. Opposite patterns

were found between species at the northern locations regarding the response to February temperature. Negative correlations with summer temperature were found in *B. pendula* populations (June at Riofrío and August at ES01). Rear-edge populations of both species positively responded to summer (July and August) precipitation, and we also found positive correlations in Riofrío with previous September and February precipitation (Figure 3).

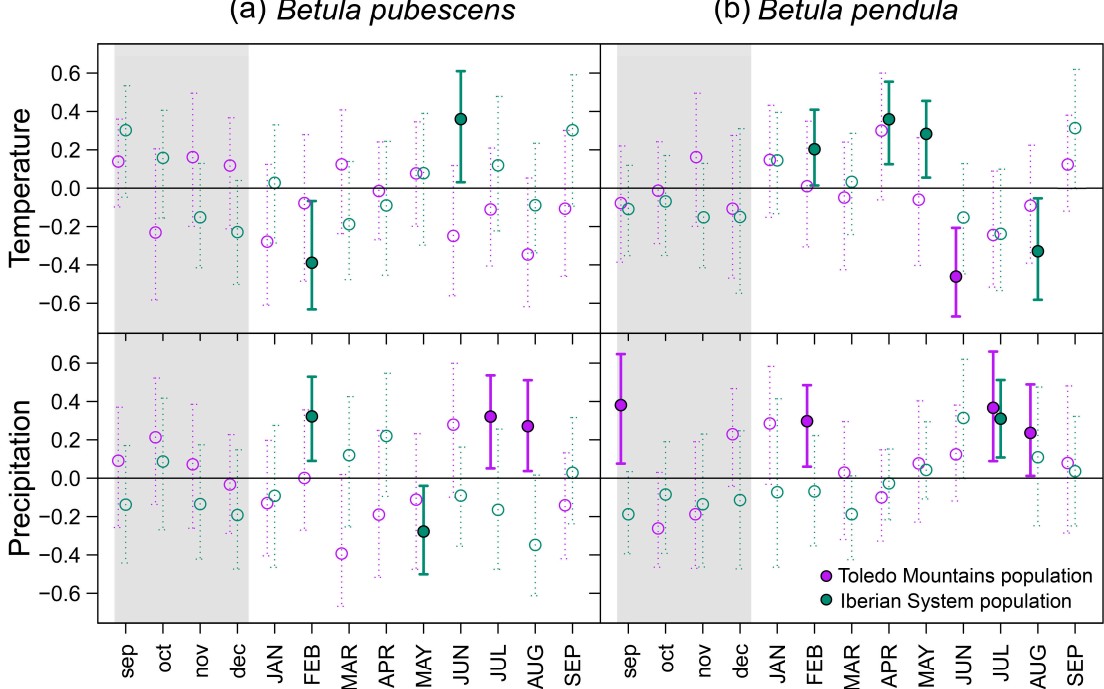

**Figure 3.** Bootstrapped correlations between monthly mean temperature (upper panels) and monthly total precipitation (bottom panels) during the period 1980–2021 based on standardized site chronologies of ring-width indices (RWI) from *Betula pubescens* (**a**) and *B. pendula* (**b**). Different colors represent rear-edge (purple symbols and bars) and core populations (purple symbols and bars). Error bars are 95%-confidence intervals, and significant and non-significant effects are represented by filled and empty symbols, respectively. Grey-shaded areas indicate the year prior to tree-ring formation (months abbreviated by lowercase letters).

These results are consistent with the correlations found with the SPEI drought index (Figure A3). The rear-edge birch populations (La Ventilla and Riofrío) and the northern population of *B. pendula* (ES01) showed the strongest correlation with SPEI3 August, that is, accumulated water deficit between June and August. Concurrently, RWI of the northern stand of *B. pubescens* (Razonzillo) showed the strongest association with SPEI3 April (February to April accumulated water deficit) (Figure A3). However, the correlation with water deficit was nonstationary in three out of the four populations studied as the effect of water deficit changed over time (Figure 4). Correlations between RWI and SPEI3 August increased over time in read-edge populations of Toledo Mountains, becoming significant since the mid and late 1990s. In contrast, *B. pubescens* growth in the Razoncillo site had a decreasing response to moisture conditions in spring. The association between summer SPEI and *B. pendula* growth in the Iberian System remained constant (Figure 4).

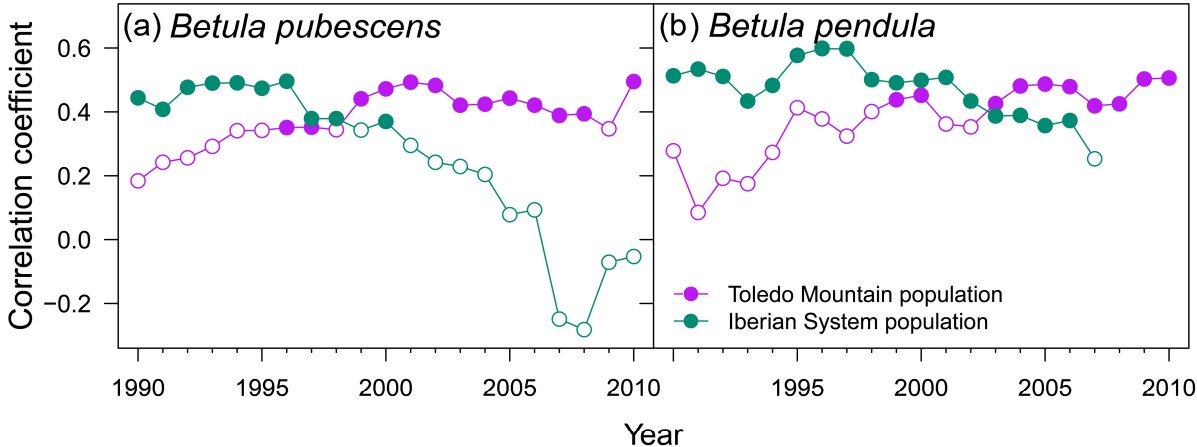

**Figure 4.** Moving bootstrapped correlation coefficients between site ring-width indices (RWI) and Standardized Precipitation-Evapotranspiration Index (SPEI) that showed the strongest correlation (see Figure A4): SPEI3 August in La Ventilla, Riofrío, and ES01, and SPEI3 April in Razoncillo. Filled and empty symbols represent significant ($p < 0.05$) and non-significant correlations, respectively.

Regarding local hydrology, we found significant correlations between growth indices and river flow data. The RWI of *B. pubescens* at the La Ventilla site was positively correlated with the water inflow to the "Torre de Abraham" reservoir from the previous November (Figure A4a), which is located upstream of the study site, and it is likely related to the feeding of the aquifer on which this stand is located. This correlation reaches higher levels of significance if only the period 2000–2019 is considered ($r = 0.563$; $p = 0.006$). *B. pendula* growth at the Riofrío site showed a positive response to Bullaque River flow during February (Figure A4b). *B. pendula* RWI in the Iberian System was negatively correlated with the flow of the Revinuesa River during the previous September (Figure A4c). We found no significant correlations between growth variability and river flow in *B. pubescens* stand in Razoncillo.

According to the LMMs, the effect of local hydrological conditions was more important than the regional climate in modulating growth variability in the *B. pubescens* stand of La Ventilla and of similar importance for the *B. pendula* stand of Riofrío (Table 2). In ES01, the effect of climate on RWI exceeded that of the river. Among the climatic variables, rear-edge populations showed a stronger impact of summer water deficit, and northern populations of the Iberian System were more sensitive to temperature (Table 2).

*3.3. Assessment of Silver Birch Growth Variability across Europe*

The GAMM characterizing the climatic impact on the growth variability of *B. pendula* across Europe included similar variables to those highlighted by the correlation analysis (Table 3). The effective degrees of freedom ranged from 2.2–2.8, indicating response curves with two inflection points (Figure 5). The nonlinear responses indicated the dependence of the responses on local climates. Summer moisture availability (SPEI3 August) had the strongest effect on RWI (Table 3). The response curve formed an arc, thus suggesting radial growth sensitivity to water deficit during June-August, which decreased and disappeared as summer moisture availability increased (Figure 5c). April temperature showed the second strongest effect on RWI, and the response curve indicated explicit sensitivity to low temperature (Temp. April < 12.5 °C) (Figure 5a). Precipitation during February limited RWI at values below 100 mm per month, while the effect of higher precipitation was not significant (Figure 5b). Finally, the response to spring moisture availability (SPEI3 May) showed a positive trend (Figure 5d), pointing out the relevance of weather conditions during the early growing season for *B. pendula* radial growth.

**Table 2.** Linear mixed effects models (LMM) characterizing the aggregate effect of regional climate and local hydrological conditions on radial growth of study birch populations in Spain during the period 1980–2021. For each predictor variable, the *F* statistic and the associated *p* value ([+] $p < 0.1$; [*] $p < 0.05$; [**] $p < 0.01$; [***] $p < 0.001$) are shown.

| Species | Site | Variable | F |
|---|---|---|---|
| *Betula pubescens* | La Ventilla | Prec. July | 2.161 |
| | | Prec. August | 0.448 |
| | | SPEI3 August | 9.317 ** |
| | | Reservoir inflow November$_{-1}$ | 15.335 *** |
| | Razoncillo | Temp. February | 1.164 |
| | | Temp. June | 10.660 ** |
| | | Prec. February | 0.216 |
| | | Prec. May | 2.780 [+] |
| | | SPEI3 April | 6.610 * |
| *Betula pendula* | Riofrío | Prec. September$_{-1}$ | 3.897 * |
| | | Prec. February | 2.472 |
| | | Prec. July | 0.539 |
| | | SPEI3 August | 7.357 ** |
| | | River flow February | 7.983 ** |
| | ES01 | Temp. February | 2.981 [+] |
| | | Temp. April | 0.317 |
| | | Temp. May | 25.944 *** |
| | | Temp. August | 29.684 *** |
| | | Prec. July | 1.760 |
| | | SPEI3 August | 4.019 * |
| | | River flow September$_{-1}$ | 3.942 * |

**Table 3.** Generalized additive mixed models (GAMM) fitting ring-width index (RWI) chronologies of *Betula pendula* across Europe against climatic variables for the period 1985–2015.

| Variable | Effective Degree of Freedom | F-Statistic | p-Value |
|---|---|---|---|
| Temp. April | 2.034 | 8.912 | 0.002 |
| Prec. February | 2.626 | 5.636 | <0.001 |
| SPEI3 August | 2.442 | 14.307 | <0.001 |
| SPEI3 May | 2.060 | 5.869 | 0.001 |
| $R^2m$ | | 0.143 | |
| $R^2c$ | | 0.303 | |

Growth variability of the southernmost rear-edge population (Riofrío) was only significantly correlated with site FR21 ($r = 0.392$, $p = 0.032$) (Figure 6a). The PC1 and PC2 accounted for 25.5% and 13.1% of the variance, respectively, and did not correlate with any geographic or climatic variable. Climate-RWI relationships of site Riofrío showed significant positive correlations with several sites from Spain (ES01, ES02), France (FR03, FR21), Switzerland (CH05, CH06), and Italy (IT07), and a negative association with a Finnish site (FI09) (Figure 6b). Accordingly, the PC2 loadings were negatively related to site latitude ($r = -0.636$, $p = 0.001$) and longitude ($r = -0.643$, $p = 0.001$), but positively related to site mean annual temperature ($r = 0.630$, $p = 0.002$) (Figure A6a) and site total annual precipitation ($r = 0.442$, $p = 0.039$). In addition, *B. pendula* RWI—SPEI3 August correlations across Europe were also negatively related to latitude ($r = -0.618$, $p = 0.002$) and longitude ($r = -0.574$, $p = 0.004$), and positively related to elevation ($r = 0.498$, $p = 0.015$) and mean annual temperature ($r = 0.517$, $p = 0.011$) (Figure A6b).

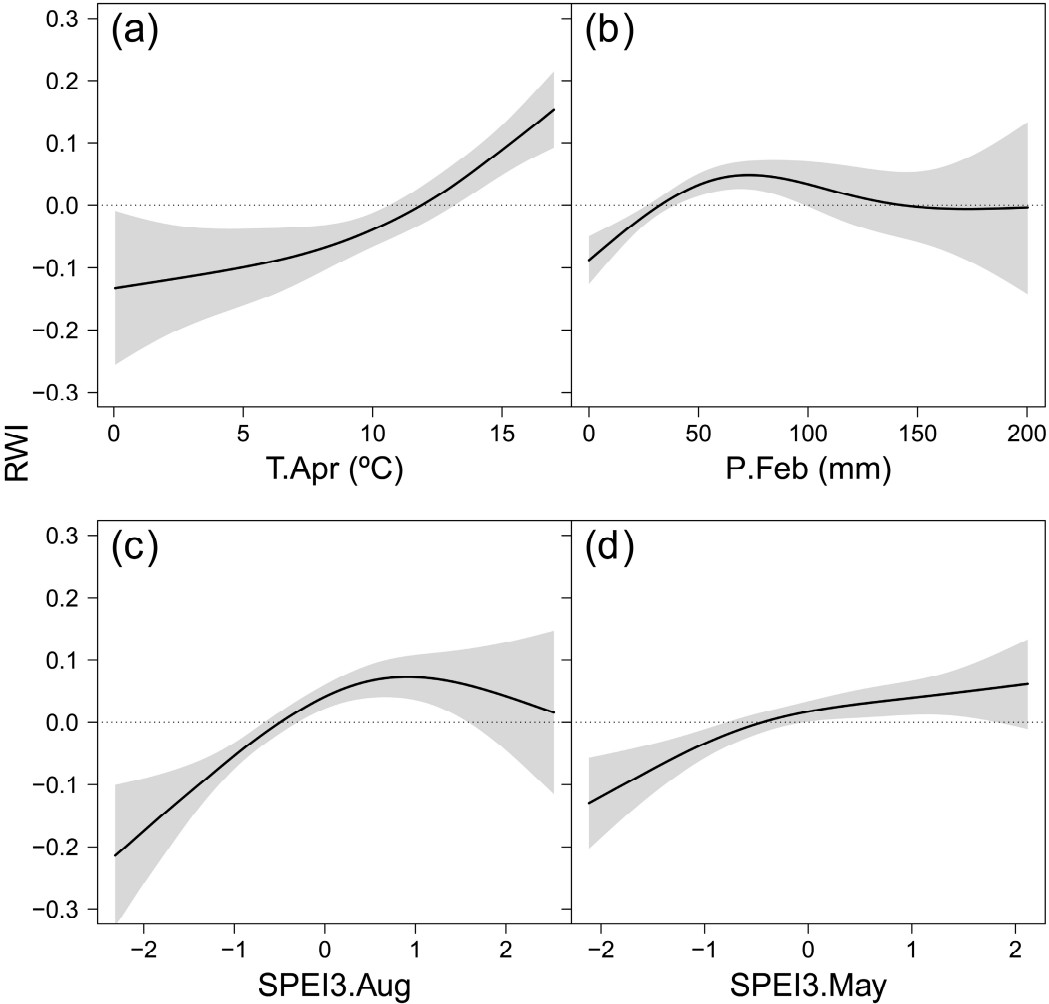

**Figure 5.** Estimated smoothing splines (black lines) and their 95% confidence intervals (shaded areas) of the responses of standardized chronologies (RWI) of *Betula pendula* to climatic variables across Europe during the period 1985–2015: Standardized Precipitation-Evapotranspiration Index of August calculated at a 3-month time scale (SPEI3.August; (**a**)), SPEI of February calculated at a 1-month timescale (SPEI1.February; (**b**)), precipitation from the previous September (**c**) and April temperature (**d**).

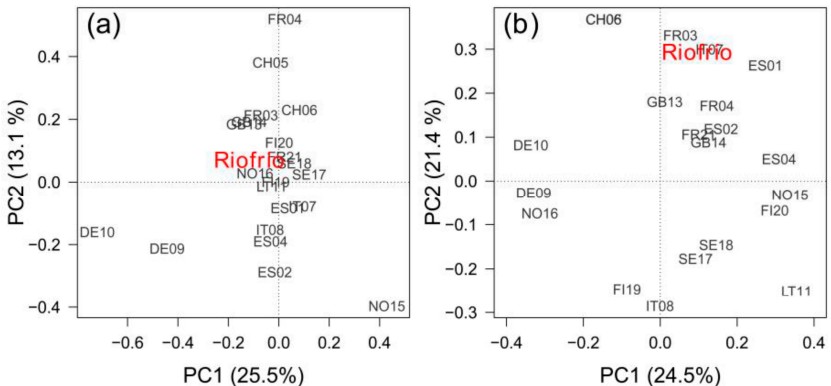

**Figure 6.** Biplots showing the sites' scores in the first (PC1) and second (PC2) principal components of a Principal Component Analysis (PCA) calculated on the covariance matrix of *Betula pendula* tree-ring width standardized chronologies (RWI) (**a**), and on the covariance matrix of correlations between monthly temperature and precipitation and RWIs (**b**) for the common period 1985–2015. The variance of each component is shown between parentheses.

## 4. Discussion

European birch species (*Betula pendula* and *B. pubescens*) display some of the widest physiological and geographical amplitudes of all Eurasian broad-leaved tree species [2,3]. In the present dendrochronological study of the two species at different spatial scales, we have found common climatic constraints on birch radial growth (Figures 3 and 5). Growth enhancement under warm conditions during the early growing season has been described previously for both species [19–21]. This is likely related to bud opening and cambium reactivation above a certain threshold of temperature sum [23,62,63]. Environmental conditions prior to the onset of stem growth also influence the interannual variability of birch growth. Such is the case of winter (February) precipitation, which we found to be positively related to radial growth (Figure 3). This effect can have two alternative explanations, as thick snow layers may prevent root freezing during winter [19], and water supply from snowmelt and winter precipitation can maintain the soil water reserve at the beginning of growth [62]. It is surprising the negative association of *B. pubescens* with February temperature (Figure 3). It could be due to the chilling requirement for the dormancy release of buds [64], which is consistent with the more cold-adapted behavior suggested for this species [24]. Nonetheless, the strongest climatic influence on the radial growth of studied birch populations was exerted by summer (June–August) water deficit at the different spatial scales analyzed (Tables 2 and 3; Figure A3). Premature leaf shedding, mortality of fine roots, and cessation of cambium activity have been related to water shortage in birch trees [62,65,66], thus reducing the growing season length and the annual tree growth. Soil water shortage has also been found to reduce leaf area and stomatal conductance, which impair photosynthesis and growth of *B. pendula* seedlings [25,66,67].

*B. pubescens* is considered more sensitive to drought and high temperatures than *B. pendula* [24]; however, our data do not seem to confirm this pattern in the comparison of Spanish birch populations located at the southern limit of their geographical distribution. This result is likely explained by the large effect of site conditions rather than regional climate on radial growth, as suggested by the significant correlations found with local hydrological conditions (Figure A4). We found a negative correlation between growth and Revinuesa River flow during the previous September in the Iberian system population of *B. pendula*. This result can be related to stress related to humid and cloudy conditions reducing carbohydrate synthesis or bud formation during the prior autumn or to the reproductive effort triggered by favorable conditions in the preceding growing season that reduces radial growth the next year [68,69].

The effect of local hydrology on rear-edge birch populations of the Toledo Mountains was similar to or even higher than that of the regional climate (Table 2). At Riofrío, dry winter and warm summer conditions likely reduce the quantity of runoff contribution to river discharge (Figure A5a), thus reducing the soil moisture available in the banks of the stream where birches grow. It is remarkable that birch growth in this site depends on previous winter precipitation (Figure 3b), which can recharge soil moisture levels to enhance radial growth in spring [36]. By contrast, we did not find significant associations between reservoir inflow and regional climate (Figure A5b), which may be related to the nearby agricultural fields that are irrigated with water not only from the reservoir but also from the aquifer that feeds the mire on which La Ventilla stand is located. Hence, climatic-driven soil moisture shortage is aggravated by human land uses at La Ventilla, which reduces groundwater availability. Indeed, one of the major threats to mires in the Toledo Mountains is the intensive drainage for watering and agricultural uses [30,40].

Long-term low growth rates and negative growth trends have been identified as early warning signals of forest dieback [70,71]. Therefore, declining growth trends shown by birch rear-edge populations that started 15–20 years ago (Figure 2) suggest their loss of vitality and the onset of canopy dieback. Considering their increasing reliance on summer water balance (Figure 4), these patterns are likely induced by the ongoing aridification of the climate of the Toledo Mountains region (Figure A2) and aggravated by the decreasing water supplied by local hydrology. Mortality events of *B. pubescens* and *B. pendula* induced

by extreme weather events have been described [15,24], as well as growth decline as a result of aridification [16,22]. For instance, [15] found that very low summer precipitation leads to low soil moisture levels and causes *B. pendula* dieback and extensive mortality in southeastern Siberia, particularly in southern and south-facing slopes with relatively drier and warmer conditions. Likewise, [16] reported a warming-induced growth decline of Himalayan birch (*Betula utilis* D. Don) in a semi-arid (southwest-facing slope) treeline in the Himalayas (central Nepal).

Nonlinear climatic responses of *B. pendula* along Europe suggest local adaptation (Figure 5), which is supported by the variability in climate-growth relationships in the different populations (Figure 6b). Growth patterns and climate-growth responses of the Riofrío rear-edge population showed similarity with the populations geographically closer and, therefore, probably present more similar climatic conditions. Indeed, we found that climatic responses followed geographical and climatic gradients (Figure A6), so that the higher the mean annual temperature, the greater the sensitivity to water deficit during summer. Great intraspecific variability in drought tolerance has been found among *B. pendula* provenances or ecotypes [67,72,73]. These studies highlighted that genotypes of dry origin perform better against drought than genotypes of wet origin, as pointed out by their lower leaf turgor loss point [72], larger osmotic adjustment, and higher stomatal conductance [67]. The isolation of birch populations in the central-southern Spanish mountains, which might have persisted here as glacial relicts, has led to their taxonomic differentiation and local adaptation to warmer and drier climates [28]. Though local adaptation can reduce the adaptability of genotypes to rapid environmental changes [33,74], the rear-edge population can provide drought-tolerant genotypes to more central distributions considering predicted species range shifts [37,75]. However, these populations face multiple threats, including drastic habitat fragmentation, warmer and drier conditions, or reduction of water supply due to human land uses. In fact, *B. pendula* subsp. *fontqueri* has been included in the red list of plants of the IUCN [76]. There is, therefore, a case for management efforts to help preserve these rear-edge populations.

Ecological studies in the study area provide several recommendations for the management and conservation of these rare Mediterranean woodlands, including (1) restoring birch stands since past and recent land-use intensification caused their decline through the direct removal of trees or the indirect drying of soils, (2) reducing the elevated density of wild ungulates which graze on birch seedlings, and (3) protecting or enlarging sensitive and vulnerable habitats such as *Sphagnum* bogs and riparian forests [30,31,77–80]. Preserving and restoring some of these threatened and protected sites, such as La Ventilla, by recharging and maintaining their soil water pools, particularly during the dry summer, is also highly recommended.

## 5. Conclusions

The assessment of radial growth patterns of rear-edge birch populations of central-southern Spain through dendroecology provides evidence of the strong effect of local hydrological conditions on growth variability, being even greater than that of regional climate in the case of *B. pubescens* populations growing on a mire. Both rear-edge birch populations showed declining growth trends that started at the beginning of the 21st century and which were not found in northern Spain. The declining growth is likely related to the increasing reliance on water summer availability, along with decreasing water supply due to climate change and agricultural land uses. The assessment of the radial growth of *B. pendula* across Europe suggested local adaptations to climate conditions. Long-term isolated, rear-edge birch populations can provide drought-tolerant genotypes to more central distributions, so management actions are required to ensure their persistence by preserving adequate ecohydrological conditions and avoiding local soil drying, which triggers dieback and increases mortality rates.

**Author Contributions:** Conceptualization, R.L.-L., J.A.L.-S. and J.J.C.; methodology, E.G.d.A., M.C. and J.J.C.; formal analysis, E.G.d.A.; investigation, J.J.C. and J.A.L.-S.; resources, J.J.C. and J.A.L.-S.; data curation, E.G.d.A. and J.J.C.; writing—original draft preparation, E.G.d.A. and J.J.C.; writing—review and editing, M.C., R.L.-L., J.A.L.-S. and J.J.C.; funding acquisition, J.J.C. All authors have read and agreed to the published version of the manuscript.

**Funding:** This research was funded by Spanish Ministry of Science and Innovation (TED2021-129770B-C21 project). E.G.d.A. was supported by CSIC (PIE-20223AT003).

**Data Availability Statement:** The data presented in this study are available on request from the corresponding author.

**Acknowledgments:** We thank the "Cabañeros" National Park personal and director for field permissions and help in La Ventilla site. We also thank "Junta de Castilla La Mancha" forest guards for their help to sampled Riofrío site.

**Conflicts of Interest:** The authors declare no conflict of interest.

## Appendix A

**Table A1.** Geographic, climatic and dendrochronological characteristics of *Betula pendula* populations distributed along a north-south gradient in Europe including field data and GenTree Dendroecological Collection [41]. Abbreviations: mean annual temperature (MAT), mean annual precipitation (MAP), annual water balance expressed as precipitation minus potential evapotranspiration (P-PET), diameter at breast height (DBH), mena tree-ring width (TRW), mean inter-series correlation of detrended tree-ring width (*Rbar*), Expressed Population Signal (EPS).

| ID | Country | Latitude (°N) | Longitude (°E) | Elevation (m a.s.l.) | MAT (°C) | MAP (mm) | P-PET (mm) | DBH (cm) | TRW (mm) | Tree Age (Years) | *Rbar* | EPS |
|---|---|---|---|---|---|---|---|---|---|---|---|---|
| FI19 | Finland | 66.37 | 26.73 | 127.13 | 0.79 | 585 | 52 | 19.94 | 1.82 | 48.2 | 0.30 | 0.923 |
| SE17 | Sweden | 64.76 | 18.70 | 278.12 | 1.66 | 578 | 45 | 22.67 | 2.39 | 40.5 | 0.32 | 0.93 |
| SE18 | Sweden | 63.92 | 19.89 | 150.95 | 3.29 | 672 | 123 | 24.11 | 2.42 | 43.5 | 0.25 | 0.887 |
| FI20 | Finland | 61.66 | 29.29 | 89.71 | 4.04 | 614 | 50 | 21.17 | 2.16 | 45.6 | 0.25 | 0.858 |
| NO16 | Norway | 60.80 | 10.75 | 187.28 | 4.56 | 667 | 105 | 25.76 | 3.81 | 27.2 | 0.32 | 0.922 |
| NO15 | Norway | 59.82 | 11.03 | 261.29 | 5.35 | 937 | 369 | 26.06 | 0.75 | 70.6 | 0.22 | 0.893 |
| GB14 | United Kingdom | 57.23 | −3.67 | 279.87 | 7.21 | 903 | 318 | 23.71 | 1.99 | 49.5 | 0.12 | 0.804 |
| LT11 | Lithuania | 55.03 | 23.01 | 56.21 | 7.42 | 632 | 15 | 22.17 | 2.12 | 25.5 | 0.32 | 0.911 |
| LT12 | Lithuania | 54.62 | 24.23 | 112.78 | 7.18 | 637 | 28 | 21.82 | 4.70 | 17.7 | 0.48 | 0.953 |
| GB13 | United Kingdom | 54.23 | −3.03 | 15.37 | 9.13 | 1450 | 825 | 30.24 | 1.87 | 62.4 | 0.15 | 0.827 |
| DE10 | Germany | 52.54 | 14.05 | 57.91 | 9.41 | 564 | −95 | 18.81 | 2.21 | 47.8 | 0.36 | 0.942 |
| DE09 | Germany | 51.84 | 14.44 | 61.95 | 9.75 | 565 | −101 | 30.00 | 2.62 | 61.2 | 0.14 | 0.809 |
| CH05 | Switzerland | 46.32 | 8.96 | 772.51 | 8.56 | 1452 | 843 | 32.28 | 2.92 | 51.1 | 0.19 | 0.842 |
| CH06 | Switzerland | 46.14 | 8.99 | 990.73 | 11.05 | 1487 | 801 | 28.33 | 2.48 | 41.4 | 0.08 | 0.691 |
| FR04 | France | 44.84 | 3.32 | 995.65 | 8.14 | 875 | 288 | 31.53 | 2.54 | 55.0 | 0.24 | 0.895 |
| FR21 | France | 44.21 | 7.10 | 1690.52 | 4.86 | 984 | 489 | 18.61 | 1.43 | 49.1 | 0.09 | 0.51 |
| FR03 | France | 44.20 | 7.07 | 1417.37 | 8.25 | 874 | 271 | 20.73 | 3.44 | 31.3 | 0.23 | 0.923 |
| IT07 | Italy | 43.61 | 11.71 | 1051.84 | 11.92 | 1150 | 458 | 26.13 | 2.01 | 56.1 | 0.05 | 0.497 |
| ES04 | Spain | 42.69 | −0.28 | 1790.00 | 6.56 | 1251 | 716 | 29.80 | 3.44 | 40.0 | 0.30 | 0.628 |
| ES02 | Spain | 42.67 | −0.32 | 1063.39 | 8.19 | 1165 | 579 | 20.74 | 2.69 | 34.9 | 0.20 | 0.845 |
| IT08 | Italy | 42.16 | 13.62 | 1552.42 | 9.81 | 1089 | 459 | 16.72 | 1.03 | 54.7 | 0.16 | 0.741 |
| ES01 | Spain | 41.97 | −2.62 | 1287.29 | 9.22 | 601 | −11 | 29.83 | 3.98 | 40.2 | 0.26 | 0.921 |
| Riofrío | Spain | 39.08 | −4.50 | 648.00 | 15.95 | 415 | −450 | 40.77 | 3.10 | 45.7 | 0.23 | 0.93 |

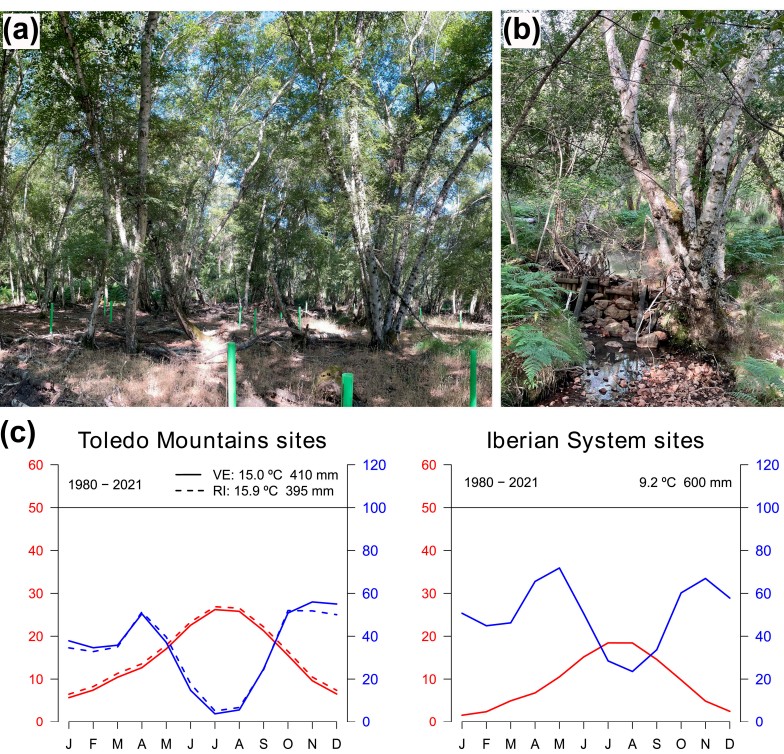

**Figure A1.** Appearance of the study stands and trees at the southernmost (rear-edge) population of *Betula pubescens* (La Ventilla; (**a**)) and *B. pendula* (Riofrío; (**b**)). Climate diagrams of the Spanish study sites at Toledo Mountains (La Ventilla solid lines; Riofrío dash lines) and Iberian System (**c**).

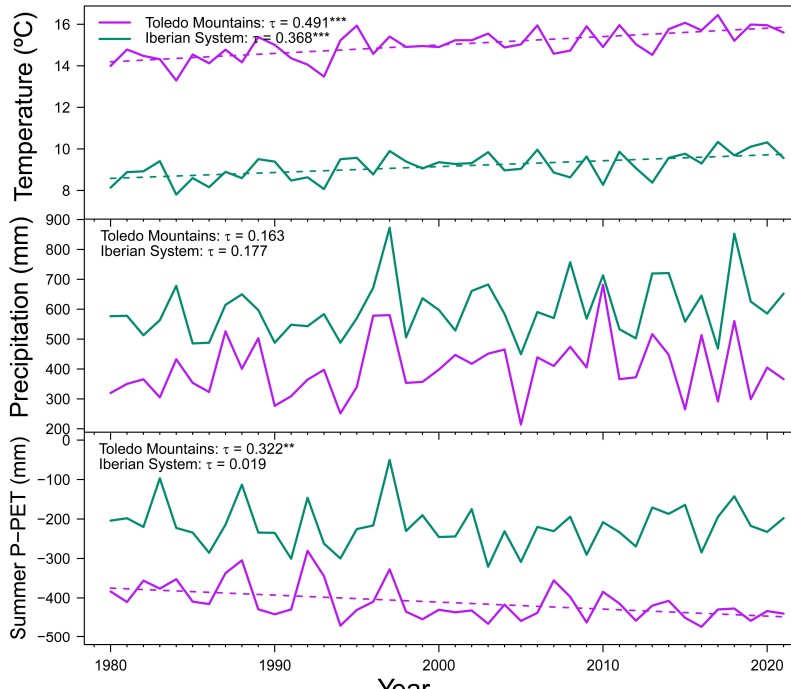

**Figure A2.** Mean annual temperature, total annual precipitation and summer water balance (precipitation minus potential evapotranspiration; P-PET) during the 1980–2021 period in the Toledo Mountains (purple lines) and Iberian System (green lines). Dash lines show significant ($p < 0.05$) trends according to Kendall $\tau$ statistic (indicated in upper left corner). Dashed lines show significant trends (** $p < 0.01$; *** $p < 0.001$).

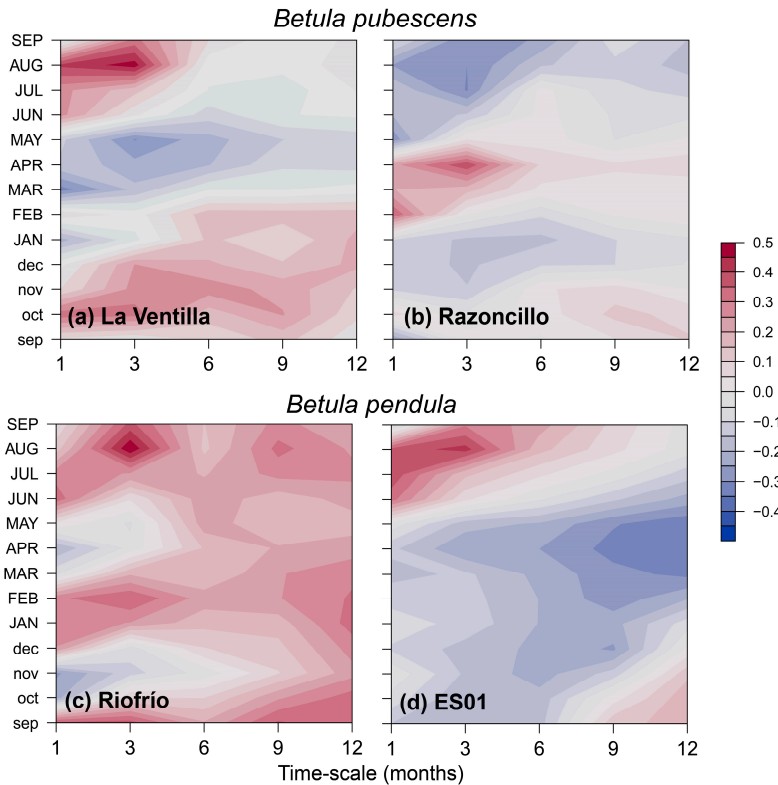

**Figure A3.** Bootstrapped correlation coefficients calculated between standardized site chronologies of tree-ring width (RWI) and the Standardized Precipitation and Evapotranspiration Index (SPEI) calculated at different time scales. The window of analysis spanned from previous September (lowercase letters) to September of the year of tree-ring formation (uppercase letters).

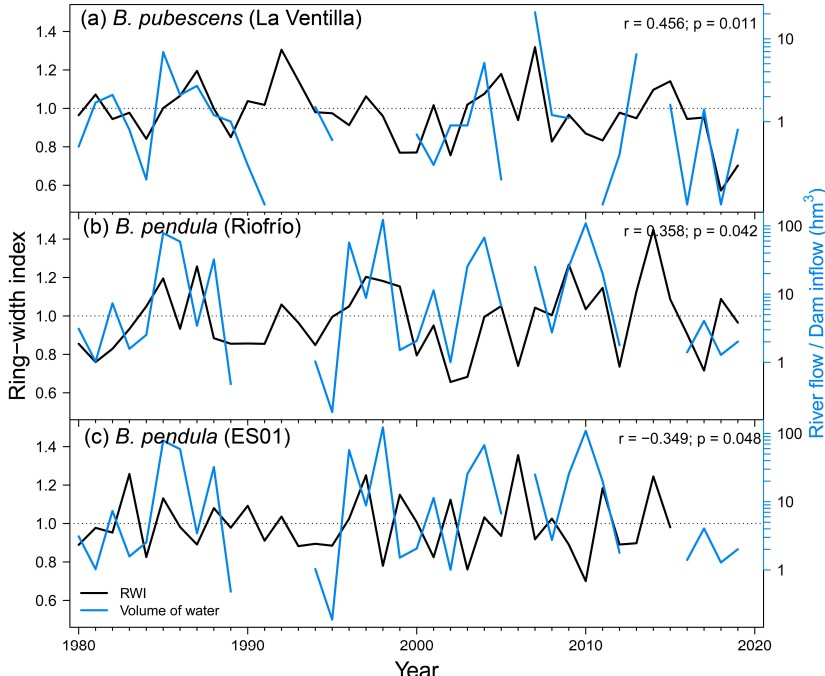

**Figure A4.** Relationship between ring-width index of birch at La Ventilla (**a**), Riofrío (**b**) and ES01 (**c**) study sites with November inflows of "Torre de Abraham" reservoir, February flow of the Bullaque River and previous September flow of Revinuesa River, respectively. Pearson correlation statistics are shown in the upper-right corner of each graph. Black and blue lines represent radial growth and water flow, respectively. Note the logarithmic scales of monthly flow and inflow data.

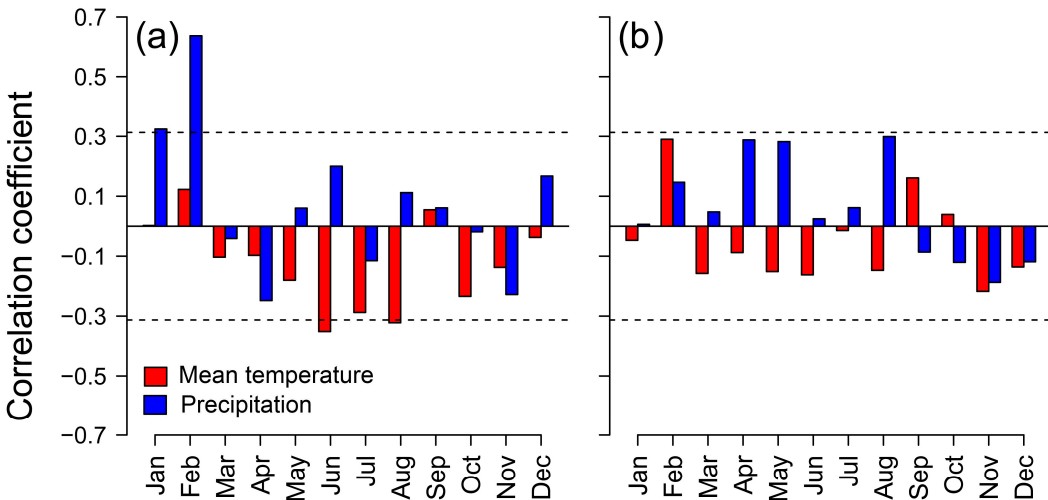

**Figure A5.** Pearson correlation coefficients of Bullaque River flow in February (**a**) and "Torre de Abraham" reservoir inflow in November (**b**) with monthly regional climatic variables (E-OBS v 27.0e dataset [43]): mean temperature (red bars) and precipitation (blue bars). Horizontal dashed lines indicate significant ($p < 0.05$) correlations.

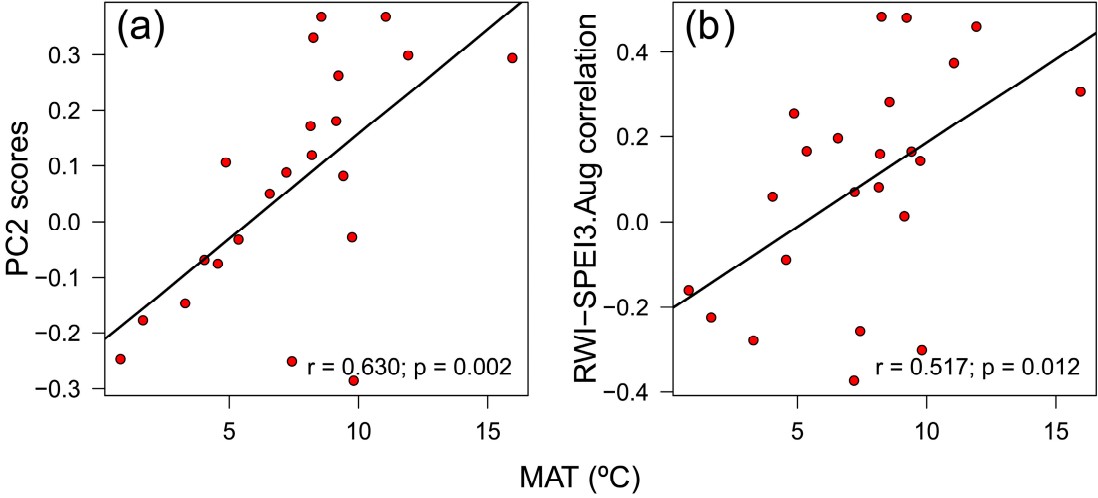

**Figure A6.** Relationships between mean annual temperature (MAT) with the PC2 scores of the Principal Component Analysis on the covariance matrix of RWI—climate correlations (**a**) and the RWI-SPEI3.Aug correlations (**b**) of *Betula pendula* across Europe. RWI is the ring-width index.

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
