# Peer review of "Sensitivity of Eurasian Rear-Edge Birch Populations to Regional Climate and Local Hydrological Conditions"

_forests, doi:10.3390/f14071360_

Round 1

Reviewer 1 Report (Previous Reviewer 1)

I have checked all three documents and I am impressed about the changes in the ms. I accept the new version.

Reviewer 2 Report (Previous Reviewer 2)

After the author's efforts to modify, although there are still some flaws in the details to be improved, I think the author can do it. As a result, the manuscript has been greatly improved and I propose to accept this manuscript for publication in Forests.

The language expression is clear and logical, and it is better to improve it by the author himself.

This manuscript is a resubmission of an earlier submission. The following is a list of the peer review reports and author responses from that submission.

Round 1

Reviewer 2 Report

This study compared the response of two birch populations to regional climatic factors and the relationship between tree rings width and nearby river and dam flow (land use) variability in the Toledo Mountains. The results found that human land uses can accelerate radial growth decline. Overall, this study is interesting and innovative, and also yields good conclusions that provide a theoretical basis for forest sustainability and scientific land use. But there have many problems and suggestions for the current manuscript. Taken all, the current manuscript should be major revised before being accepted for publication in the Forests.

I have the following major concerns should be considered: 1) The author wanted to demonstrate tree radial growth more vulnerable to local water draining than to regional climate warming in the title. However, it was found that drought-induced growth decline can be ‘accelerated’ by local water draining. Therefore, considering ‘More vulnerable’ is inappropriate in the title. 2) The study should demonstrate the relationship between radial growth response to regional climate variability and water draining with full results, thus illustrating the obvious constraints of water draining on birch’s growth. First, only the relationship analysis of temperature and precipitation is not sufficient to explain drought constraints (Drought-induced growth decline). Second, the study should consider how to link regional climate change, tree growth and water draining to clearly illustrate the reason of radial growth decline, rather than the correlation of tree rings with the other both. 3) I see that the tree decline trend started to be significant after 2000, especially for Betuta pubescens, why is the statistical analysis starting year 1960, including Figure 5? 4) Manuscript study of two Birch species, but agricultural land uses only could exacerbate drought stress of one of the populations (La Ventilla, Betuta pubescens) in the conclusion. So what about the Riofrío species (B. pendula)? This is inconsistent with the conclusion of the title. 5) Consider that the decline trend of Betuta pubescens starts after 2000, when the precipitation decreasing and the temperature increasing trend are more significant. But what is the relationship between Betuta pubescens and November reservoir inflows after 2000, suggesting that the analysis is more significant? The results are not visible at the moment in Figure 5. 6) It is necessary to consider the effect of August maximum temperature on Betuta pubescens. It is not unusual that the current Moving correlation appears almost insignificant for Betuta pubescens.

Detailed comments are as follows:

1. Abstract

1) L14, It is suggested that ‘but’ and ‘at the same time’ should not be used together.

2) L20-21, ‘a decreasing limitation by cold spring temperature and an increasing constraint by summer precipitation’. Is this the result of moving correlation? Meanwhile, this result seems to be less applicable to both birch species.

2. Materials and methods

3) L99, ‘two birch populations’. Suggest writing the names of two birch species. Such as (La Ventilla, Betuta pubescens subsp. celtiberica) and (Riofrío, B. pendula subsp. fontiqueri).

4) L117, The climate conditions in the study area need to be presented in an appendix, not just a text description.

5) L165, Why did you choose a cubic regression spline de-trending method and on what basis? Also, the reference on which it is based should be cited. Does the de-trending method need to be selected according to the riverine environment at the sampling site?

6) L172-177, Add citation.

7) L196, Why moving bootstrapped correlations considering moving 25-year long. I have seen many studies choosing 30-year long. Suggested Citation.

3. Results

8) L208, I see lines 154 TRW stands for tree ring width. The meaning of the abbreviation should be uniform throughout the text.

9) L257, Does ‘water inflow’ refer to river flow?

Figure 5, Suggestions for labeling color lines represent what.

5. Discussion

10) L285, ‘Soil water shortage’. Soil moisture and hydrological conditions of tree growth relationships are close and further analysis is recommended.

6. Conclusion

11) The conclusions suggest further summary generalization of the results, and the current conclusions confirm that Betuta pubescens at La Ventilla is more affected by hydrological conditions, but the Title says that birch populations are all more vulnerable to local water draining.

7. Figures and Tables

12) Figure 1, Map elements are incomplete, pointers or latitude and longitude need to be marked.

13) Table 1, The longitude of the two sampling sites should be in a uniform format. Such as ‘4°’ and ‘04°’.

14) Table 1, Why Pteridium aquilinum showed in Table.

15) Table 1, DBH and TRW are mean DBH and mean TRW in the label of Table 1? And, TRW is calculated from the residual chronologies? It should be stated.

Reviewer 3 Report

This paper compared two rear-edge birches populations inhabiting two different ecosystems in the Toledo Mountains, and assessed radial growth patterns through dendroecological approaches and examined the effects of temperature and precipitation. The theme of this paper is very valuable, which may provide a reference for the adequate management to protect the endangered rear-edge birch populations. The manuscript is well-written and easy to understand. In particular, the discussion and analysis are very detailed and powerful.

The necessary modifications and improvements are as follows:

1. L37, L41: Please show the authors of the cited papers.

2. Table 1: The longitude of Riofiro should be “4º 30’ 10” W”.

3. L211: The positive trend seems had disappeared in about 1970th in Riofrio according to the figure 2. How did you determine the breakpoint was 2001?

4. L355-357: The relationship between growth and precipitation seems more significant for birches at Riofrío (Figure 3, 4). Why the conclusions state drought stress of the population in La Ventilla leading birch growth to more negetive trends?

Reviewer 4 Report

This concise and clear paper considers the impact of climate and land-use changes upon birch populations in Spain. It presents very interesting and valuable results which undoubtedly deserve to be published in the Forests journal. Yet there are a couple of points of minor importance which the authors may wish to consider before the publication:

Line 100 – while authors provided short description of the vegetation of the second site, they did not conduct the same for the first one. Would not it better to describe vegetation of the mire as well?

Line 119 – are there meteorological stations close to the sampling sites? If yes, why climatic data were retrieved from the gridded CRU dataset instead of using direct measurements? It seems this point has to be clarified

Line 134 – are there any quantitative measure of the mire drying like measurements of the changes of the water table?

Line 141 – was it line INTERCEPT method? In any case a bit more detailed description of the method is needed

Line 151 – may be trunk is more acceptable term than foot?

Line 162 – BAI was calculated for individual series of tree-ring measurements. However two radii were measured for each tree. Is not it better to calculate mean tree-ring widths for each tree first and then BAI for a tree?